# Moderate-Intensity Intermittent Training Alters the DNA Methylation Pattern of PDE4D Gene in Hippocampus to Improve the Ability of Spatial Learning and Memory in Aging Rats Reduced by D-Galactose

**DOI:** 10.3390/brainsci13030422

**Published:** 2023-02-28

**Authors:** Jinmei Zhang, Qiaojing Gao, Jun Gao, Liting Lv, Renfan Liu, Yi Wu, Xue Li, Yu Jin, Lu Wang

**Affiliations:** 1School of Sports Medicine and Health, Chengdu Sport University, Chengdu 610041, China; 2School of Physical Education, Chaohu University, Hefei 238000, China

**Keywords:** aging, moderate-intensity intermittent training, hippocampus, PDE4, learning and memory

## Abstract

(1) Background: Aging is the main risk factor for most neurodegenerative diseases, and the inhibition of Phosphodiesterase 4(PDE4) is considered a potential target for the treatment of neurological diseases. The purpose of this study was to investigate the inhibitory effect of moderate-intensity intermittent training (MIIT) on PDE4 in the hippocampus of rats with D-galactose (D-gal)-induced cognitive impairment, and the possible mechanism of improving spatial learning and memory. (2) Methods: the aging rats were treated with D-Gal (150 mg/kg/day, for 6 weeks). The aging rats were treated with MIIT for exercise intervention (45 min/day, 5 days/week, for 8 weeks). The Morris water maze test was performed before and after MIIT to evaluate the spatial learning and memory ability, then to observe the synaptic ultrastructure of the hippocampus CA1 region, to detect the expression of synaptic-related protein synaptophysin (SYP) and postsynaptic density protein 95 (PSD95), and to detect the expression of PDE4 subtypes, cAMP, and its signal pathway protein kinase A (PKA)/cAMP response element binding protein (CREB)/brain-derived neurotrophic factor (BDNF), and the PDE4 methylation level. (3) Results: we found that MIIT for 8 weeks alleviated the decline in spatial learning and memory ability, and improved the synaptic structure of the hippocampus and the expression of synaptic protein SYP and PSD95 in D-Gal aging rats. To elucidate the mechanism of MIIT, we analyzed the expression of PDE4 isoforms PDE4A/PDE4B/PDE4D, cAMP, and the signaling pathway PKA/CREB/BDNF, which play an important role in memory consolidation and maintenance. The results showed that 8 weeks of MIIT significantly up-regulated cAMP, PKA, p-CREB, and BDNF protein expression, and down-regulated PDE4D mRNA and protein expression. Methylation analysis of the PDE4D gene showed that several CG sites in the promoter and exon1 regions were significantly up-regulated. (4) Conclusions: MIIT can improve the synaptic structure of the hippocampus CA1 area and improve the spatial learning and memory ability of aging rats, which may be related to the specific regulation of the PDE4D gene methylation level and inhibition of PDE4D expression.

## 1. Introduction

Aging is a major risk factor for most neurodegenerative diseases. Cognitive decline occurs with age, both in normal aging and in pathological conditions [1,2]. The hippocampus is a key brain region for regulating learning and memory function, and has attracted much attention in the research on aging and cognition [3]. Although the neurobiological mechanism of age-related learning and memory function has not yet been fully elucidated, it is clear that the decline in age-related spatial learning and memory ability is related to bilateral hippocampus synaptic damage [4]. The integrity and plasticity of the synaptic structure affect the information transmission between neurons, and play an important and positive role in the formation of learning and memory [5].

Phosphodiesterase 4 (PDE4) is an enzyme that catalyzes the hydrolysis of cyclic adenosine monophosphate (cAMP) and plays an important role in memory consolidation and maintenance [6]. Inhibition of PDE4 is considered a potential target for the treatment of nervous system diseases. Several studies have shown that the inhibition of PDE4 can improve hippocampus long-term potentiation (LTP) and dendritic spine density by activating cAMP/protein kinase A (PKA)/cAMP response element binding protein (CREB)/brain-derived neurotrophic factor (BDNF) signaling, increase synaptic plasticity, and then improve cognitive function [7,8,9]. For example, inhibition of PDE4 by the first-generation inhibitor Rolipram increased the level of cAMP and p-CREB in the hippocampus, as well as the survival and proliferation of new neurons in the hippocampal dentate gyrus, significantly improving hippocampal long-term potentiation [10,11,12]. Later, several research groups reported different types of small molecule PDE4 inhibitors, such as roflumilast, apremilast, etc. [13,14,15]. However, despite the broad therapeutic potential of these small molecule PDE4 inhibitors in preclinical studies, the progress of most molecules in the clinic has been hampered, possibly due to severe side effects such as headache, diarrhea, and dizziness [7,16,17]. Therefore, it is an ideal strategy to find a non-side-effect inhibitory pathway targeting PDE4 to repair the age-related deficit of synaptic plasticity and the decline in learning and memory ability.

In rodent and human studies, exercise has been shown to prevent or delay the onset of neurodegenerative diseases [18]. Exercise can regulate the expression of BDNF, promote the plasticity of brain structure and function, and improve cognitive function [19]. Previous study has shown that aerobic exercise can down-regulate the expression of PDE4 and improve brain function of aging rats [20], but the mechanism is not clear. In recent years, high-intensity interval training (HIIT) has attracted more and more attention as a time-efficient exercise. HIIT is characterized by short periods of high-intensity exercise alternating with rest or low-intensity exercise. Similar or greater mental and physical health benefits can be obtained with reduced time investment compared to traditional moderate to high-intensity continuous exercise [21,22,23]. Studies have shown that moderate intensity interval training (MIIT) and HIIT have similar training effects in cardiac and metabolic studies [24,25,26,27], but MIIT is more suitable for the elderly or groups with low activity capacity [28]; MIIT is relatively rare in cognition-related studies.

Here, in order to explore the effect of MIIT on PDE4 expression and its role in the recovery of cognitive aging, we selected the D-galactose (D-Gal)-induced aging model, which is a systematic and homogeneous aging model of comprehensively accelerated aging and cognitive deficits [29,30,31]. Eight weeks of MIIT intervention on D-gal-induced aging rats showed that 8 weeks of MIIT inhibited the expression of PDE4D protein by specifically regulating the methylation of PDE4D, activated the up-regulation of BDNF expression mediated by the cAMP/PKA/CREB signaling mechanism, and then improved the synaptic structure and functional plasticity of the hippocampus of aging rats, and improved the spatial learning and memory ability of aging rats. Our study provides a promising strategy to slow down age-related cognitive aging.

## 2. Materials and Methods

### 2.1. Animals and Study Design

Thirty-nine male Sprague-Dawley (SD) rats (8 weeks of age, weighing 220 ± 20 g at the beginning of the experiment) were purchased from Chengdu Dashuo Biological Technology Co., Ltd. (Chengdu, China) (Certification No: SCXK (chuan) 2020-030). All rats were housed in plastic containers under standard laboratory environments (temperature (22 ± 1 °C), humidity (60 ± 10 °C), 12 h light/dark cycle, and were given free access to food and water). All animal studies were performed per the “Principles of Laboratory Animal Care” (National Institutes of Health publication number 80–23, revised in 1996) and were approved by the Animal Care and Use Guidelines Committee of Chengdu Sport University (approval number: No. 2015-018).

Rats were randomly divided into three experimental groups as follows: Control group: young group treated with saline (*n* = 13). D-Gal Group: aging group induced by D-galactose, *n* = 13). D-Gal+E Group: aging group induced by D-galactose that performed MIIT (*n* = 13). After establishing the aging model, the rats in each group were trained and tested by water maze for the first time, and then Control group and D-Gal group were fed naturally, and D-Gal+E group was given MIIT for 8 weeks. The experimental procedure is shown in Figure 1a.

### 2.2. Establishment of the Aging Model

D-galactose (D-Gal) (IG0540) was purchased from Beijing Solarbio Science & Technology Co., Ltd. (Beijing, China). The model of Aging was established by injecting D-Gal (150 mg/kg/day) dissolved in saline (0.9% NaCl) intraperitoneally for 6 weeks, as described in previous studies [32]. The same volume of saline was injected in Control group during this period.

### 2.3. Exercise Protocols 

The protocol of the MIIT was adapted from previous studies [33]. An animal electric treadmill machine (DSPT-208, Shanghai Xinruan Information Technology Co., Ltd., Shanghai, China) was used in this study. Rats in D-Gal+E groups were subjected to a familiarization period of treadmill exercise (10 min/day; the treadmill speed was 10 m/min) of 1 week. After the adaptation period, rats performed the MIIT. The details of the MIIT protocol were as follows: 10 m/min for 10 min (warming-up), after the warm-up exercise, 20 m/min for 4 min, 12 m/min for 3 min, and 5 groups were repeated alternately. Finally, the treadmill for 45 min/day was completed. The slope of the treadmill was 0°. 

### 2.4. Morris Water Maze Behavioral Test

Morris water maze (MWM) experiment is divided into two stages: “positioning navigation experiment” and “spatial exploration experiment” were used to evaluate hippocampus-dependent spatial learning and memory capacity of D-gal-induced aging rats as previously reported [34]. The water maze (ZH, Anhui Zhenghua Biological Instrument Co., Ltd., Suixi, China) was a circular open swimming arena (diameter 150 cm, height 40 cm) filled with opaque water at 23 ± 1 °C, under dim light and in a room containing external cues. A platform was submerged for 2 cm underwater in one of four identical quadrants. Before MIIT, the rat was placed in the maze to swim freely for 2 min, and if the rat did not find the platform within 2 min, the rat was guided to stay on the stage for 10 s to let it know that there was an escape platform in the maze, and to try to make it remember the approximate position of the platform. Then, the MWM was administered daily over a period of 4 days. Rats were released into water at different starting points and subjected to three trials each day to find the platform. Rats that failed to find the platform within 2 min were gently guided to the platform and allowed to stay on the platform for 10 s. The escape latency that rats spent in finding the platform and the swimming speed and the swimming distance were recorded. Subsequently, the space exploration experiments were performed 24 h after the positioning navigation test ended. The platform was removed and the rat was placed in one-third of the pool. Rats were allowed to swim freely for 1 min. The number of platform crossings, target quadrant swim time ratio, and target quadrant swim distance were recorded. After 8 weeks of aerobic intermittent training, a 2-day positioning navigation experiment was conducted, and on the third day, a spatial exploration experiment was conducted, and the experimental method was the same as before the movement. A camera (WV-CP500D, Panasonic, Kadoma, Japan) above the tank was used to record the swimming track of the animal. Behavioral data were recorded and analyzed by a computer installed with a visual analyzing system (ZH, Anhui Zhenghua Biological Instrument Co., Ltd., Suixi, China). 

### 2.5. Tissue Preparation

Twenty-four hours after the end of MWM, rats were deeply anesthetized with Pentobarbital sodium and then decapitated, craniotomy and rapid removal of brain tissue, stripped of the hippocampus, washed with normal saline and quickly frozen in liquid nitrogen, and stored at −80 °C for further analysis.

### 2.6. Transmission Electron Microscopy Assay

Hippocampus samples of 3 rats in each group were collected after being perfused transcardially with 200–300 ml of 3% glutaraldehyde. Immersion fixation was then completed at a size of around 1 mm3. Samples were placed in 3% glutaraldehyde at 4 °C for 2 h. After being post-fixed with 1% aqueous osmium tetroxide in a 0.2 M cacodylate buffer (pH 7.2) for 2 h, then cleaning, dehydration, embedding, and solidification. An ultra-thin slicer was used to prepare ultra-thin tissue sections with a thickness of about 50 nm, then float on the groove liquid surface, and remove the slices to the copper mesh, 3% uranium acetate-lead citrate double staining, transmission electron microscope observation, photography, and observed with the JEM-1400PLUS at 120 kV (JEOL, Inc., Tokyo, Japan). Ultrastructural changes in the hippocampus synapses were observed under TEM (HT7700-SS, Hitachi, Tokyo, Japan). Bouton parameters were quantified using Image J version 1.42 software (National Institutes of Health, Bethesda, MD, USA).

### 2.7. ELISA

The hippocampus tissue was fully homogenized in precooled 0.01 m phosphate-buffered saline (pH = 7.4) and centrifuged at 12,000× *g* at 4 °C; the supernatant was collected. The concentration of cAMP was determined by the RatCampELISAKIT kit (Sambega SBJ-R0140; Nanjing, China). The optical density at 450 nm was measured by enzyme labeling instrument. Three multiple holes are made for each sample to reduce the test error.

### 2.8. Gene expression Analysis 

Total RNA was extracted from hippocampus tissue, and the hippocampus tissue of all rats using a Molpure^®^ Cell/Tissue Total RNA Kit (YEASEN, Shanghai, China) and then quantified with a NanoDrop one spectrophotometer (Nanodrop Technologies, Wilmington, DE, USA). Reverse transcription of RNA into cDNA was conducted using the Hifair^®^ III 1st Strand cDNA Synthesis Kit (gDNA digester plus) (YEASEN, Shanghai, China). RT-PCR was carried out with Hieff^®^ qPCR SYBR^®^ Green Master Mix (YEASEN, Shanghai, China), and all reactions were conducted in triplicate. The mRNA levels were normalized to Gapdh mRNA levels and quantified using the 2^−ΔΔCt^ method. All samples were analyzed in duplicate. All primer sequences are listed in Table 1.

### 2.9. Western Blot Analysis

The hippocampus was homogenized in RIPA lysis buffer (add 1% complete protease inhibitors and 2% phosphatase inhibitor cocktail; Beyotime, Shanghai, China). Tissue lysates were mixed with 5× SDS-PAGE loading buffer (Beyotime), loaded on a 5–12% SDS polyacrylamide gradient gel, and subsequently transferred onto a polyvinylidene fluoride. The blots were blocked in 5% BSA in Tris-Buffered Saline with Tween (TBST) and incubated with rabbit anti-PDE4A antibody (DF 13317, 1:500, Affinity Biosciences Cincinnati, OH, USA), rabbit anti-PDE4B antibody (72096, 1:1000, CST), rabbit anti-PDE4D antibody (DF6535, 1:800, Affinity Biosciences Cincinnati, OH, USA), rabbit anti-PKA antibody (AF5450, 1:800, Affinity Biosciences Cincinnati, OH, USA), rabbit anti-CREB antibody (9197, 1:1000, CST), rabbit anti-p-CREB antibody (9198, 1:800, CST), rabbit anti-BDNF antibody (DF6387, 1:700, Affinity Biosciences Cincinnati, OH, USA), and rabbit anti-GAPDH antibody (5174, 1:1000, CST). Horseradish peroxidase-conjugated Goat anti-Rabbit IgG secondary antibodies (ZB-2301, 1:5000, ZSGB-BIO) and a Clarity ECL kit (P0018S, Beyotime) were used to detect protein signals. Multiple exposures were taken to select images within the dynamic range of the film. Selected films were scanned (300 dpi) and quantified using Image J version 1.42 software (National Institutes of Health, Bethesda, MD, USA). GAPDH bands were used for normalization.

### 2.10. DNA Methylation Analysis

Genomic DNA was extracted from the hippocampus CA1 region tissue collected, using the Genomic DNA extraction kit (Qiagen DNeasy kit, 69506, Hilden, Germany) was used to extract DNA, the QiagenEpiTect Bisulfite Kit (Qiagen, 59104, Hilden, Germany) was used for bisulfite conversion. Polymerase chain reaction (PCR) primers and pyrosequencing primers (Table 2) were designed using PyroMark Assay Design 2.0 software (Qiagen, Valencia, CA, USA) to target six CpG sites in the promoter region and 12 CpG sites in the exon-1 region of the PDE4D gene. The promoter region was defined as the gene region directly upstream 2 kbp range of the transcription start site (TSS). PCR reactions were performed with KAPA DNA polymerase reagents (Kapa Biosystems, kk2101, USA). PCR was performed under the following conditions: 95 °C, 3 min; 94 °C, 30 s; 40 cycles of 56 °C, 30 s; 72 °C, 1 min; 72 °C, 7 min; and 4 °C hold. The PCR product was sequenced using PyroMark Q96 (Qiagen, PyroMark Q96 ID, Valencia, CA, USA) Pyro sequencer. Epi-Tect high and low methylated control DNA (Qiagen Sciences, Germantown, MD, USA) were included with every pyrosequencing experiment.

### 2.11. Statistical Analysis

All data were analyzed using SPSS version 25 (IBM corporation, Armonk, NY, USA). Statistical significance was defined as a *p* value less than 0.05. The data were shown as the mean ± standard error of the mean (SEM). GraphPad Prism 8.0 software (GraphPad Software Inc., SanDiego, CA, USA) was used to draw all graphs. 

## 3. Results

### 3.1. Eight Weeks MIIT Improves Spatial Learning and Memory in D-Gal-Induced Aging Rats 

The MWM experiment was designed to evaluate the effect of 8-week MIIT exercise on the improvement of cognitive impairment in D-Gal-induced aging rats. Before MIIT exercise, rats in each group were trained for 4 consecutive days in a positioning navigation experiment and 1 day in a spatial exploration experiment test to evaluate the influence of D-Gal injection on the spatial learning and memory ability of rats. The average swimming speed on the first day showed no significant difference among the three groups (Figure 1b), suggesting that D-Gal did not change the average swimming speed of D-Gal group and D-Gal+E group. For the average escape latency of each group of rats for locating the underwater escape platform from 1 to 4 days before exercise intervention, with the passage of time, all rats’ time for finding the underwater escape platform shortened, indicating that all groups completed the spatial learning process in the acquisition process. On the 1st day and the 4th day, the mean latencies of D-Gal group and D-Gal+E group were significantly higher than those of Control group (D-Gal group vs. Control group, *p* < 0.05; D-Gal+E group vs. Control group, *p* < 0.05, Figure 1c), showed slower learning speed. The accuracy of long-term memory was evaluated in the space exploration experiment. One-way ANOVA showed that D-Gal group and D-Gal+E group had fewer times of crossing the platform compared with Control group (D-Gal group vs. Control group, *p* < 0.05; D-Gal+E group vs. Control group, *p* < 0.05, Figure 1d). There was no significant difference in the average latency and the number of crossing platforms between D-Gal and Dal+E groups before exercise intervention (*p* < 0.05, Figure 1d). After 8 weeks of MIIT, the positioning navigation experiment and space exploration experiment were carried out for 2 days again. The average swimming speed on the first day of the positioning cruise showed that the swimming speed of aging rats was improved by exercise training (D-Gal group vs. D-Gal+E group, *t* = 2.539, *p* < 0.05) (Figure 1e). In order to avoid the influence on escape latency caused by the difference in swimming speed, the learning speed and memory ability of rats were evaluated by the swimming distance and target quadrant distance before going on stage instead of the escape latency and the number of crossing platforms. The two days positioning navigation experiment showed that D-Gal group had to swim a longer distance to find the platform than Control group, and the average swimming distance of the D-Gal+E group was shorter than that of D-Gal group on the first day, but there was no statistical difference (*t* = 2.037, *p* = 0.056). On the second day, compared with D-Gal group, the swimming distance before the stage in Control group was significantly shorter (*t* = 2.059, *p* < 0.05), while the swimming distance before the stage in D-Gal+E group was significantly shorter than that in D-Gal group (*t* = 2.856, *p* < 0.01) (Figure 1f). The swimming distance of D-Gal+E group was significantly higher than that of D-Gal group (*t* = 2.411, *p* < 0.05). The results showed that 8 weeks of MIIT significantly improved the spatial learning and memory ability of aging rats.

### 3.2. Eight Weeks MIIT Improves the Ultrastructure of Hippocampus Synapses and Up-Regulates the Expression of Synaptic Associated Proteins in Aging Rats

The synaptic ultrastructure was observed by transmission electron microscopy, and the expression of synaptic-related protein PSD95 and SYP protein was detected by Western blot. The results of transmission electron microscope observation showed that compared with Control group, the average synaptic density in the hippocampus CA1 region of aging rats induced by D-Gal decreased significantly (*t* = 6.118, *p* < 0.01, Figure 2a,b), the thickness of postsynaptic density decreased significantly (*t* = 5.257, *p* < 0.01, Figure 2a,c), and the synaptic gap increased significantly (*t* = 9.319, *p* < 0.01, Figure 2a,d). Compared with D-Gal group, the synaptic density was significantly increased (*t* = 4.031, *p* < 0.01, Figure 2a,b), the thickness of postsynaptic density was significantly increased (*t* = 6.359, *p* < 0.01, Figure 2a,c), and the synaptic cleft was significantly decreased (*t* = 10.778, *p* < 0.01, Figure 2a,d) in D-Gal+E group. Western blot showed that the protein expressions of PSD95 and SYP in D-Gal group were significantly down-regulated compared with those in Control group (*t* = 2.877, *p* < 0.05, Figure 2e; *t* = 7.314, *p* < 0.01, Figure 2f). Compared with D-Gal group, the expression of PSD95 and SYP protein in the hippocampus of D-Gal+E group was significantly up-regulated (*t* = 10.975, *p* < 0.01, Figure 2e; *t* = 3.079, *p* < 0.05, Figure 2f).

### 3.3. MIIT Promotes cAMP/PKA/CREB/BDNF Signaling Pathway up-Regulation in Aging Rats

To study the effect of MIIT on the cAMP/PKA/CREB/BDNF signaling pathway in aging rats, we used ELISA to detect cAMP content in the hippocampus of rats in three groups, and Western blot to detect the protein expression of PKA, CREB\P-CREB, and BDNF (Figure 3). Compared with Control group, the cAMP content in D-Gal group was significantly decreased (*t* = 4.827, *p* < 0.01), and the protein expression of PKA/p-CREB/BDNF was down-regulated (*t* = 2.614, *p* = 0.059, Figure 3b,c; *t* = 2.959, *p* < 0.05, Figure 3b,d; *t* = 8.275, *p* < 0.01, Figure 3b,e). Compared with D-Gal, 8 weeks of MIIT could effectively reverse the decrease in cAMP content in aging rats (*t* = 2.886, *p* < 0.05), and up-regulate the expression of PKA/p-CREB /BDNF protein (*t* = 6.148, *p* < 0.01, Figure 3 b,c; *t* = 8.199, *p* < 0.01, Figure 3b,d; *t* = 8.954, *p* < 0.01, Figure 3b,e).

### 3.4. MIIT Effectively Reduced PDE4D Expression but Not PDE4A and PDE4B

PDE4 is a major cAMP-degrading enzyme. We examined the mRNA and protein expression of three major isoforms, PDE4A, PDE4B, and PDE4D. Compared with Control group, the expression of PDE4 mRNA in the hippocampus of D-Gal group was higher, but PDE4A and PDE 4B had no significant difference, and PDE4D was significantly higher (*t* = −2.538, *p* < 0.05, Figure 4a). Compared with D-Gal group, the expression of PDE4D mRNA in D-Gal+E group was only significantly downregulated (*t* = 2.538, *p* < 0.05), while the expression of PDE4A and PDE4B mRNA had no significant change. PDE4A, PDE4B, and PDE4D protein levels in D-Gal group were significantly higher than those in Control group (*t* = 2.653, *p* = 0.057, Figure 4b,c; *t* = 3.702, *p* < 0.05, Figure 4b,e; *t* = 4.615, *p* < 0.05, Figure 4b,f). However, 8 weeks of MIIT could significantly down-regulate the expression of PDE4D protein in the hippocampus of aging rats (*t* = 4.615, *p* < 0.05), while there was no significant change in the levels of PDE4A and PDE4B protein.

### 3.5. MIIT Regulates Methylation Levels in PDE4D Promoter and Exon1 Regions

Next, we examined the effect of 8 weeks of MIIT on the methylation of the CG site in the PDE 4D DNA promoter region and exon1 region of the rats’ hippocampus by pyrosequencing. The methylation level of different CG sites is quite different, and there is a high methylation level (>90%) at promoter site 1 and Exon 1 site 2 (Figure 5). Compared with Control group, the methylation level of promoter site 2 (*t* = 2.240, *p* < 0.01) and Exon 1 site 4 CG (*t* = 3.421, *p* < 0.05) in D-Gal group was significantly decreased. Compared with D-Gal group, the methylation levels of promoter site 2 (*t* = −2.802, *p* < 0.05), promoter site 4 (*t* = −2.555, *p* < 0.05), and exon 1 site 4 CG (*t* = −3.344, *p* < 0.05) were significantly increased after 8 weeks of MIIT. These results suggest that 8 weeks of MIIT can modulate the epigenetic status of the PDE 4D promoter and exon 1 regions in the hippocampus, and that these effects may be the mechanism by which exercise reduces PDE4D expression.

## 4. Discussion

Using senescence-accelerating model organisms is an important method to study the cellular and molecular mechanisms of hippocampal aging and cognitive decline. Long-term administration of high doses of D-galactose induces memory and cognitive impairment in rodents [31,35,36]. Superoxide anion produced by D-galactose metabolism directly damages tissues and organs, causes oxidative damage and neuroinflammation to the brain [29,37,38], further damages the plasticity of synaptic structure and function of the hippocampus, and finally leads to neurodegenerative symptoms [39]. Therefore, the aging model induced by D-galactose can be used to simulate the brain aging process and evaluate the effect of anti-aging research. In this study, we used the MWM task to evaluate the spatial learning and memory ability of rats. The results showed that the spatial learning and memory ability of rats in an aging model group decreased after 6 weeks of D-Gal intervention.

In this study, 8 weeks of MIIT significantly improved the swimming speed of aging rats. In the orientation cruise test, 8 weeks of MIIT was beneficial to the retention of long-term memory. In the post-exercise spatial exploration test, the motor performance of D-Gal+E rats proved that 8 weeks of MIIT effectively reversed the damage to memory ability caused by aging. Previous studies have shown that exercise is beneficial to brain function, and can prevent or delay the occurrence of neurodegenerative diseases related to aging. In animal models, exercise can regulate the expression of ganglionic brain-derived neurotrophic factor, promote brain structural and functional plasticity and hippocampal neurogenesis, and improve cognitive function [40]. The plasticity of hippocampal synaptic structure is an important basis for functional plasticity and the cellular basis of learning and memory [41]. The plasticity of synaptic structure is manifested by the changes in synaptic number and size, synaptic active area length, synaptic cleft width, synaptic curvature, and postsynaptic density thickness [42]. We found that 8 weeks of MIIT intervention significantly increased the synaptic number and postsynaptic density thickness in the hippocampal CA1 area of aging rats. The width of the synaptic cleft in the hippocampus of aging rats was shortened. The thickness of the postsynaptic density reflects the postsynaptic neuronal mechanism of synaptic structural plasticity [43]. The synaptic cleft is responsible for neurotransmitter transmission between pre-synaptic neurons and post-synaptic neurons, and optimal shortening of the synaptic cleft may have the function of optimizing synaptic strength [44]. Synaptic protein SYP is the marker protein of presynaptic vesicles of nerve cells [45], and PSD95 is the most important and abundant protein on the postsynaptic membrane, which is necessary for the activity and stability of postsynaptic membrane receptors [46]. Studies have shown that synaptic proteins provide structural integrity to protect synaptic transmission and are extremely closely linked to learning and memory capabilities. Excessive loss of synaptic proteins may lead to cognitive impairment [46,47]. Our study showed that both SYP and PSD95 were down-regulated in the hippocampus of D-Gal-induced aging rats and recovered after 8 weeks of MIIT. Therefore, the changes in synaptic number, synaptic structural parameters, and the increase in the SYP\PSD95 level induced by 8 weeks of MIIT lead to the increase in neurotransmitter release, thus delaying the decline of spatial learning and memory ability of aging rats. Therefore, 8 weeks of MIIT induced an increase in synaptic number, changes in synaptic structural parameters, and up-regulation of hippocampal SYP and PSD95 levels, thus delaying the decline of spatial learning and memory ability in aging rats.

BDNF plays an important role in the plasticity of neural circuits in mature brain [48]. The cAMP cascade plays an important role in neuronal plasticity and memory function. Increased levels of cAMP activate PKA, which, in turn, phosphorylates CREB, and p-CREB binds to the cAMP response element (CRE), which initiates the encoding of BDNF and regulates synaptic plasticity by upregulating BDNF [49]. It has been shown that the cAMP signal transduction system is disrupted in the aging-related neurodegenerative brain, and the disruption of cAMP signal transduction is responsible for memory impairment and neuronal loss [50]. Similar changes were observed in the present study, where 6 weeks D-Gal administration resulted in a significant down-regulation of cAMP levels of the hippocampus, as well as a significant decrease in the levels of its downstream signaling protein PKA and p-CREB. Eight weeks of MIIT can significantly up-regulate cAMP and PKA levels in the hippocampus of aging rats, increase the phosphorylation of CREB, and also significantly up-regulate BDNF. Therefore, 8 weeks of MIIT may increase the expression of endogenous BDNF in the hippocampus by up-regulating the level of cAMP/PKA/p-CREB, thus improving the learning and memory ability of aging rats.

PDE4 specifically hydrolyzes cAMP and plays an important role in regulating learning and memory. The inhibition of PDE4 can effectively improve the decline of age-related cognitive function. PDE4 has four subtypes, PDE4A, PDE4B, PDE4C, and PDE4D. Each subtype can hydrolyze cAMP, but PDE4A, PDE4B, and PDE4D have high expression in the brain and have neuroprotective effects after inhibition [51]. Therefore, we examined the mRNA and protein expression levels of PDE 4A\B\D. We found that PDE 4B/D increased significantly in the hippocampus of the D-Gal aging model, but PDE4A did not change significantly. It has been reported that changes in PDE4B concentration are associated with depression and anxiety-like behaviors [52], while PDE4A may play a neuroprotective role by modulating neuroinflammation [53]. Although some studies have shown that exercise can improve the inflammatory response and reduce the depression-like behavior of aging rats [54]; however, MIIT did not significantly alter PDE4A/PDE4B expression in this study, so it may not have been through the PDE4A/PDE4B pathway. PDE4D is mainly expressed in the hippocampus and plays an important role in hydrolyzing cAMP [55]. Our results show that 8 weeks of MIIT effectively inhibits PDE4D mRNA and protein over-expression, suggesting that 8 weeks of MIIT up-regulates cAMP/PKA/p-CREB levels and increases endogenous BDNF expression in the hippocampus, which is associated with the inhibition of PDE4D expression.

DNA methylation is one of the epigenetic mechanisms that control gene expression after exercise [56], and methylated DNA can inhibit transcription by interfering with the binding of transcription factors and by assembling transcription mechanisms into regulatory sites of genes [57]. Increasing evidence suggests that DNA methylation is involved in the effects of exercise on the brain [58], and exercise can differentially induce changes in DNA methylation to regulate synaptic plasticity and learning and memory, such as whereby exercise can positively regulate the expression of memory consolidation genes (such as BDNF and CREB) through DNA methylation [59], and can also down-regulate genes that have an inhibitory effect in these events [38,60]. Studies on the role of DNA methylation on the regulation of gene expression have focused on the transcription initiation site (TSS), and it has been found that the methylation density of promoter and exon 1 is negatively correlated with the corresponding gene expression level [60]. Six CG sites were found in the TSS −2000 bp region and 12 CG sites were found in the exon1 (230 bp) when the PDE4D gene sequence was queried. We examined the methylation levels of CG sites in the PDE4D DNA promoter region (6) and Exon1 region (12) using pyrosequencing. Significant methylation reduction was found at the second site in the promoter region and the fourth site in the exon region in aging rats, which was reversed by 8 weeks of MIIT. The specific methylation of these CG sites may be the main mechanism of the inhibition of PDE4D gene expression by MIIT.

## 5. Conclusions

This study demonstrated the improvement of cognitive function in D-GAL-induced aging rats by 8 weeks of MIIT. MIIT can improve the spatial learning and memory ability of aging rats by changing the synaptic structure of the hippocampus. In addition, the positive effect of exercise is related to the activation of the cAMP / PKA / CREB / BDNF signal pathway. Our results also show that MIIT can upregulate the methylation level of the local CG site in the promoter region and the exon1 region of the PDE4D gene in the hippocampus, which indicates that MIIT can inhibit the expression of PDE4D and improve the spatial learning and memory of aging rats. The possible mechanism is related to the specific regulation of the methylation level of the PDE4D gene.

## Figures and Tables

**Figure 1 brainsci-13-00422-f001:**
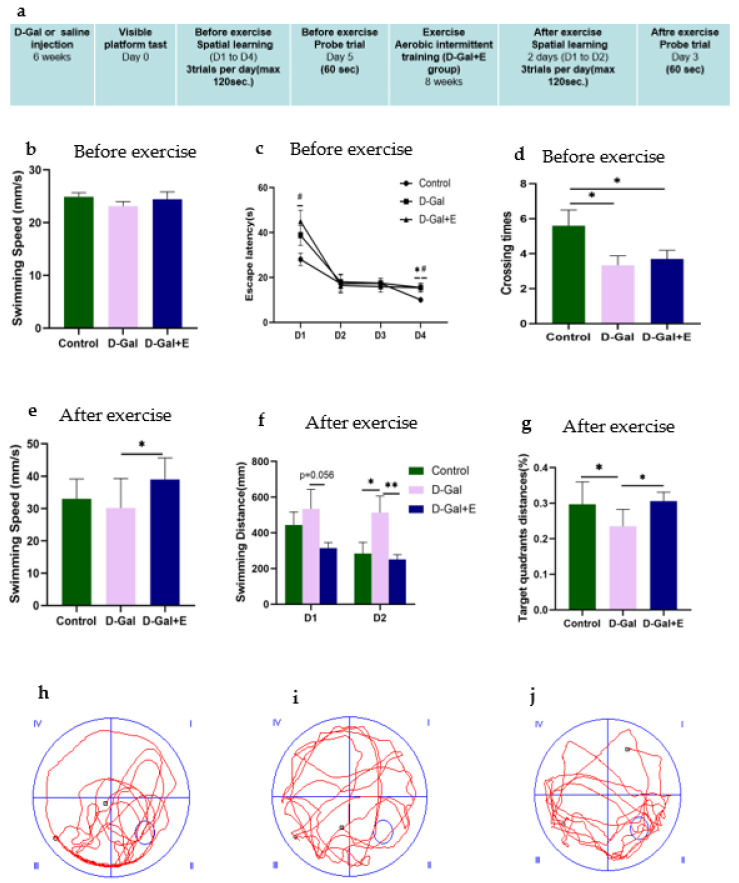
Morris water maze was used to evaluate the spatial learning and memory ability of rats: (**a**) Experimental design process and arrangement. (**b**) Average swimming speed on the first day of the positioning navigation experiment before the MIIT (mean ± SEM, *n* = 13). (**c**) The average escape latency in searching for hidden platforms in the l positioning navigation experiment (mean ± SEM, *n* = 13). (**d**) After the platform was removed, the number of crossing the original platform in 60 s was measured before MIIT (mean ± SEM, *n* = 13). (**e**) Average swimming speed on the first day of the positioning navigation experiment after MIIT (mean ± SEM, *n* = 13). (**f**) Positioning navigation experiment after MIIT, swimming distance when looking for hidden platforms (mean ± SEM, *n* = 13). (**g**) The percentage of distance in the original platform quadrant (target quadrant) within 60 s in each group of rats in the post-MIIT spatial exploration experiment (mean ± SEM, *n* = 13). (**h**–**j**) Trajectory diagram of space exploration experiment of rats in each group after MIIT. One-way ANOVA and LSD post hoc test were used for pre-MIIT data analysis, and Independent Samples *t*-Test was used for post-MIIT data analysis. * *p* < 0.05 vs. D-Gal group, ** *p* < 0.01 vs. D-Gal; ^#^
*p* < 0.05 vs. D-Gal+E group. Control: young group treated with saline, D-Gal Group: aging group induced by D-galactose, D-Gal+E Group: aging group induced by D-galactose that performed MIIT.

**Figure 2 brainsci-13-00422-f002:**
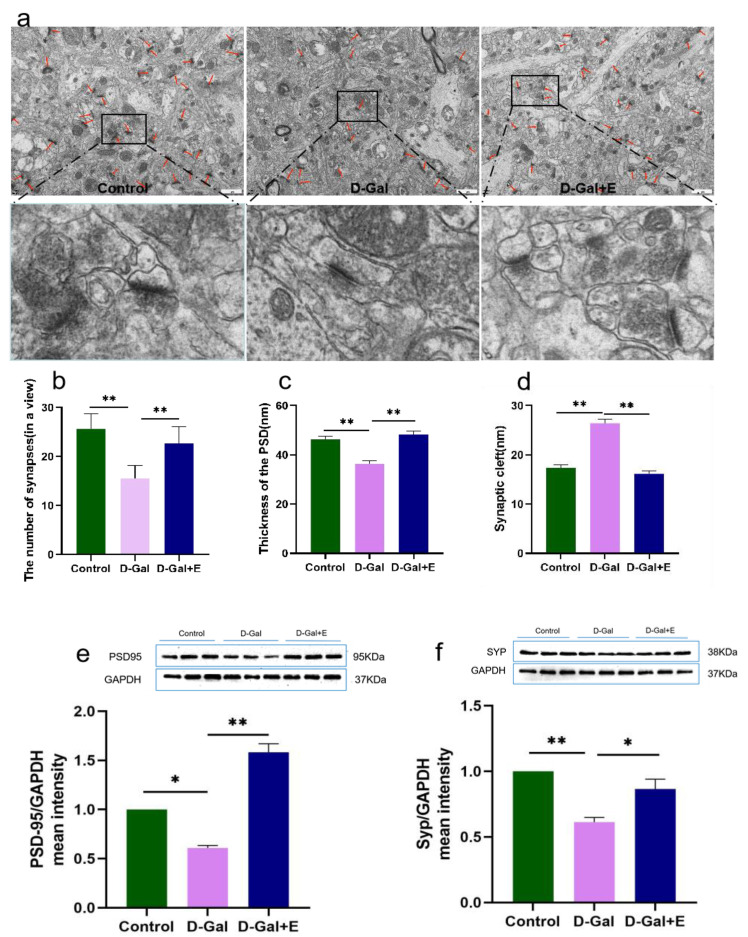
MIIT improved the ultrastructure of hippocampus synapse and up-regulated the expression of synaptic associated proteins in aging rats: (**a**) Electron microscopic images of hippocampus synapses, Scale bar = 1 µm, lower panel shows boxed areas showing higher magnification views of synapses. (**b**) Average number of synapses in a single field (mean ± SEM, *n* = 6). (**c**) Quantitative analysis of synaptic PSD thickness (mean ± SEM, *n* = 95–144). (**d**) Synaptic space width (mean ± SEM, *n* = 95–144). (**e**) Western blot analysis of hippocampus PSD95 expression levels (mean ± SEM, *n* = 3). (**f**) Western blot analysis of hippocampus SYP expression (mean ± SEM, *n* = 3). Independent Samples *t*-Test, * *p* < 0.05 vs. D-Gal group, ** *p* < 0.01 vs. D-Gal group. Control: young group treated with saline, D-Gal Group: aging group induced by D-galactose, D-Gal+E Group: aging group induced by D-galactose that performed treadmill exercise.

**Figure 3 brainsci-13-00422-f003:**
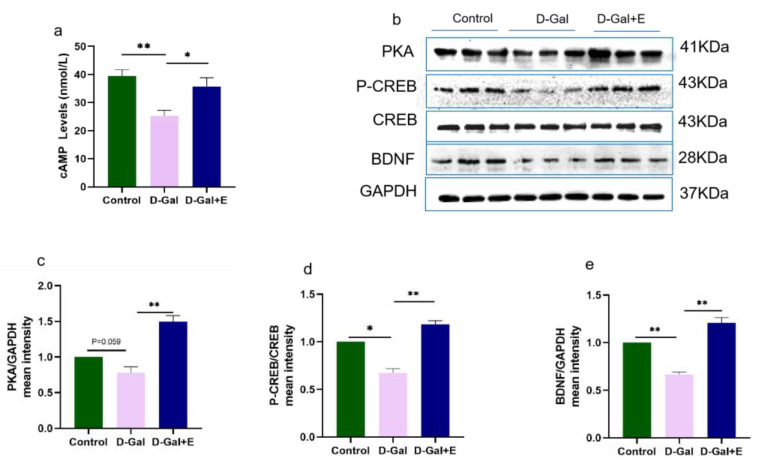
Eight weeks MIIT up-regulated the expression of cAMP and PKA/p-CREB/BDNF in hippocampus: (**a**) cAMP level in hippocampus was analyzed by Elisa. (**b**–**e**) Western blot was used to analyze the protein expression of PKA, p-CREB, and BDNF in hippocampus. The above data are presented in the form of mean ± SEM, Independent Samples *t* Test, * *p* < 0.05 vs. D-Gal group, ** *p* < 0.01 vs. D-Gal group. Control: young group treated with saline, D-Gal Group: aging group induced by D-galactose, D-Gal+E Group: aging group induced by D-galactose that performed treadmill exercise.

**Figure 4 brainsci-13-00422-f004:**
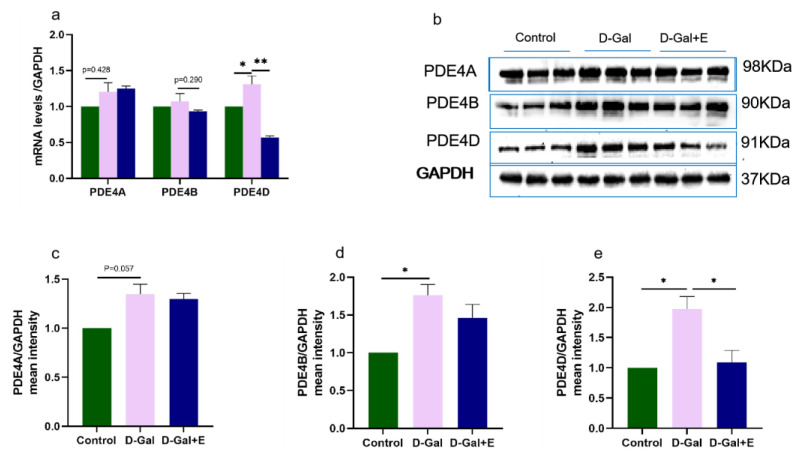
Effects of MIIT on the expression of PDE4A/B/D mRNA and protein in aging rats: (**a**) The expression of PDE4A/B/D mRNA in hippocampus was detected by Real-Time PCR (mean ± SEM, *n* = 6). (**b**–**e**) Western blot analysis of PDE4A/B/D protein expression levels (mean ± SEM, *n* = 3). Independent-Samples *t* Test, * *p* < 0.05 vs. D-Gal group, ** *p* < 0.01 vs. D-Gal group. Control: young group treated with saline, D-Gal Group: aging group induced by D-galactose, D-Gal+E Group: aging group induced by D-galactose that performed treadmill exercise.

**Figure 5 brainsci-13-00422-f005:**
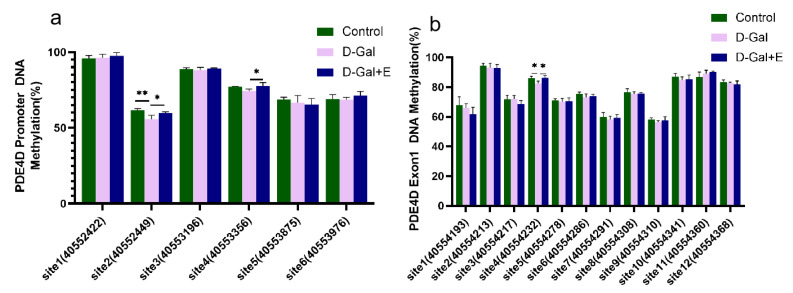
Methylation changes in CG sites in PDE4D promoter region and exon 1 region in hippocampus of rats in each group: (**a**) Methylation level of each CG site in the promoter region (mean ± SEM, *n* = 4). (**b**) Methylation level of each CG site in exon1 region (mean ± SEM, *n* = 4). Independent-Samples *t* Test, * *p* < 0.05 vs. D-Gal group, ** *p* < 0.01 vs. D-Gal group. Control: young group treated with saline, D-Gal Group: aging group induced by D-galactose, D-Gal+E Group: aging group induced by D-galactose that performed treadmill exercise.

**Table 1 brainsci-13-00422-t001:** Primers used in the Gene expression analysis.

Gene Name	Forward Primer (5′→3′)	Reverse Primer (5′→3′)
PDE4A	GGCGTCTCCAACCAGTTCCTAATC	TCTCTTCTTGCAGCAGCTTGAATC
PDE4B	GACTGGTACTTCATGCCGCCTTC	ATCAGGTCATCGCCGTGTTGTTC
PDE4D	TCTGGCGTCCTCCTCCTTGATAAC	CTTTGTGGGGTTGCTCAGGTCTG
GAPDH	GACATGCCGCCTGGAGAAAC	AGCCCAGGATGCCCTTTAGT

**Table 2 brainsci-13-00422-t002:** Primers or TaqMan probes used for gene expression and DNA—Methylation analysis.

Gene/region	Sequence 5′→3′
PDE4D promoter 1: site1(40552422), site2(40552449)	PCR primer forward: AGTTTTGATGATTGGGTTTGTAT PCR primer reverse: AACCCCTAAAAACTTATACTTCCTCTT Pyrosequencing primer: GGGTTTGTATTTTTATATATGTTAG
PDE4D promoter 2: Site3(40553196)	PCR primer forward: GAGTGGGTGTGTGGATGAGTA PCR primer reverse: AAAAAACCCCCCATCCCAA Pyrosequencing primer: AGTATTTTTGTGGAGGT
PDE4D promoter 3: Site4 (40553356)	PCR primer forward: GGGGTGTTGAGTTTTTATAGGAATAAG PCR primer reverse: ACTAAACTACTAACCAAACAAAACACATA Pyrosequencing primer: GGTTATATATGTGGTTGG
PDE4D promoter 4: Site5(40553875)	PCR primer forward: GGGAGTATTTTGTTTTTTTTTTTAAGAAGT PCR primer reverse: CCAACAAACAACATTAACATAACACTAT Pyrosequencing primer: CTAACAATCTTACAATTTACCA
PDE4D promoter 5: Site6(40553976 )	PCR primer forward: AGATTGTTAGGGATATTAATTGTAATGAT PCR primer reverse: CCAACAAACAACATTAACATAACACTAT Pyrosequencing primer: ACATAACACTATCTACTACAAAA
PDE4D Exon-1 1: Site1(40554193 ) Site2(40554213) Site3(40554217) Site4(40554232)	PCR primer forward: TTGAAGGTAAGAGGAGGAAGTTG PCR primer reverse: AACCCATACTAAAACTCCTATTAATTC Pyrosequencing primer: AGAGGAGGAAGTTGG
PDE4D Exon-1 2: Site5(40554278) Site6(40554286) Site7(40554291) Site8(40554308) Site9(40554310) Site10(40554341) Site11(40554360) Site12(40554368)	PCR primer forward: TGTGGAAGGGTTATTAAAGGTATT PCR primer reverse: ATCCAACTTCCTCCTCTTACC Pyrosequencing primer: GGGTTATTAAAGGTATTTATTTAG

## Data Availability

Datasets analyzed in the current study are available from the corresponding author on request.

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
