# Peer review of "Moderate-Intensity Intermittent Training Alters the DNA Methylation Pattern of PDE4D Gene in Hippocampus to Improve the Ability of Spatial Learning and Memory in Aging Rats Reduced by D-Galactose"

_brainsci, 2023, doi:10.3390/brainsci13030422_

Round 1
Reviewer 1 Report
The manuscript entitled “Moderate-Intensity Intermittent Training Alters the DNA Methylation pattern in PDE4D Gene in Hippocampus to improve the ability of Spatial Learning and Memory in Aging Rats” by Zhang et al. investigated the effect of moderate-intensity intermittent training (MIIT) on spatial learning and memory and on PDE4 gene in the hippocampus of aging rats. The authors reported that 8 weeks of MIIT improved learning and memory processes, synaptic structure of hippocampus and the expression of synaptic protein SYP and PSD95 in D-Gal aging rats. In addition, the results showed that MIIT can upregulate the methylation level of local CG site in the promoter region and the exon1 region of PDE4D gene in hippocampus, which may be related with the memory improvement.
On the whole, this study seems well conducted, methods are sound, data are convincing and the conclusions are in general supported by the data. However, the discussion is short and leaves a certain amount of dissatisfaction. There are a few issues, which need to be addressed:
· the text has capital letters in the middle of the sentence, e.g. “Several studies have shown that inhibition of PDE4 can improve hippocampus Long-term potentiation(LTP) and dendritic spine density by activating cAMP/ protein kinase A(PKA)/ cAMP response element binding protein(CREB)/Brain dervied neurotrophic factor(BDNF) signaling, increase synaptic plasticity, and then improve cognitive function”, please correct it
· in the introduction section the abbreviation should be entered first, line: 69-72, please correct it
· in line 74-76 there is some ambiguity in the text, please explain
· Please explain if saline was given to 3 groups?, all groups should be treated the same
· In line:111-112 please explain where this sum of 45 min comes from? Because after the summary it comes out to be 35 min
· Why is there no F value in the results, why is everything counted by the t-test?
· In the discussion the sentence in line: 436-440 is unclear, please modify it
Author Response
Thank you very much for your modification suggestions. Now according to your suggestions, the author makes the following modifications
Point1.the text has capital letters in the middle of the sentence, e.g.“Several studies have shown that inhibition of PDE4 canimprove hippocampus Long-term potentiation(LTP) and dendritic spine density by activating cAMP/ protein kinase A(PKA)/ cAMP response element binding protein(CREB)/Brain dervied neurotrophic factor(BDNF) signaling, increase synapticplasticity, and then improve cognitive function”, please correct it
Response 1:Amend as follows:Several studies have shown that inhibition of PDE4 can improve hippocampus long-term potentiation(LTP) and dendritic spine density by activating cAMP/ protein kinase A(PKA)/ cAMP response element binding protein(CREB)/brain dervied neurotrophic factor(BDNF) signaling, increase synaptic plasticity, and then improve cognitive function
Point 2. in the introduction section the abbreviation should be entered first, line: 69-72, please correct it.
Response 2:Amend as follows:In recent years, high-intensity intermittent training (HIIT) has been paid more and more attention as a time- efficient exercise. HIIT is characterized by short periods of high-intensity exercise alternating with rest or low-intensity exercise.
Point 3. in line 74-76 there is some ambiguity in the text, please explain
Response 1:Amend as follows: In recent years, high-intensity interval training (HIIT) has attracted more and more attention as a time-efficient exercise. HIIT is characterized by short periods of high-intensity exercise alternating with rest or low-intensity exercise. Similar or greater mental and physical health benefits can be obtained with reduced time investment compared to traditional moderate to high intensity continuous exercise[21-23].Studies have shown that moderate intensity interval training(MIIT) and HIIT have similar training effects in cardiac and metabolic studies [24-27], but MIIT is more suitable for the elderly or groups with low activity capacity[28], MIIT is relatively rare in cognition-related studies.
Point4. Please explain if saline was given to 3 groups?, all groups should be treated the same.
Response 4:The aging model group was injected with D-galactose, which was diluted to 7.5% concentration with saline, the injection volume was 2 ml/kg/d, and the control group was injected with the same dose of saline.
Point5. In line:111-112 please explain where this sum of 45 min comes
from? Because after the summary it comes out to be 35 min.
Response 5:Each workout also includes a 10-minute warm-up, so there are 45 minutes in total.The training schedule is as follows:
Speed (M/min) |
10 m/min
|
20 m/min
|
12 m/min
|
20 m/min
|
12 m/min
|
20 m/min
|
12 m/min
|
20 m/min
|
12 m/min
|
20 m/min
|
12 m/min
|
Time (min) |
10min |
4min |
3min |
4min |
3min |
4min |
3min |
4min |
3min |
4min |
3min |
Point6. Why is there no F value in the results, why is everything
counted by the t-test?
Response6 :For the water maze data processing before exercise intervention, because the only difference factor among the groups was aging, the one-way analysis of variance was used. In all subsequent data processing, aging and exercise were two factors among the three groups, the independent sample T- test was used for the comparison between the control group and the d-gal group, and the comparison between the d-gal+E group and the d-gal group.
Point7. In the discussion the sentence in line: 436-440 is unclear,
please modify it.
Response 7:Amend as follows:although some studies have shown that exercise can improve the inflammatory response and reduce the depression-like behavior of aging rats [54], however, MIIT did not significantly alter PDE4A /PDE4B expression in this study, it may not be through PDE4A/PDE4B pathway. PDE4D is mainly expressed in hippocampus and plays an important role in hydrolyzing cAMP[55]. Our results show that 8 weeks MIIT effectively inhibits PDE4D mRNA and protein overexpression, suggesting that 8 weeks MIIT up-regulates cAMP/ PKA/p-CREB levels and increases endogenous BDNF expression in the hippocampus, which is associated with the inhibition of PDE4D expression.

Reviewer 2 Report
Major Points:
1. The authors use treatment with D-galactose as a model for the aging brain. They provide 3 references on this approach (their references 27, 28 and 29). None of these references is from a top-100 journal. There needs to be more emphasis in the Introduction and also in the Discussion on the various aspects of the model: How does it compare to study of actual aged rats, and why was it used here instead of using aged rats or mice? The use of the term “aged rats” in the title is misleading: these are not aged rats, they are D-galactose-treated rats.
2. In their discussion of HIIT and MIIT (and the relative virtues of each), the authors cite 2 references (their references 20 and 21). These references appear to focus on the cardiac and metabolic effects of these interventions, rather than their effect on the CNS. What is the state of the literature on the effects of these interventions on the aging CNS? It appears that references 30 and 31 refer to exercise paradigms other than HIIT and MIIT.
3. There needs to be more Methods information on the Morris Water Maze. What instrumentation (video or other) was used to monitor the animals (please specify manufacturer) and what software was used to capture the data, prepare statistical analyses, and prepare the Morris Water Maze figures?
4. Why was only the Morris Water Maze test used, instead of additional tests of learning, memory and cognition?
5. There needs to be some additional information regarding the immunoblots. The authors state that PKA, PDE4A, and PDE4D antibodies were obtained from Affinity. The authors need to provide references to prior studies that validate these antibodies. Alternatively, they need to provide data/figures that validate the antibodies: e.g., compare the signal generated in cells transfected to express PKA, PDE4A, or PDE4D, respectively, with appropriate controls (e.g., vector-only transfected cells). References on prior studies that have used the other antibodies (e.g., PDE4B, CREB, pCREB, BNDF) would be highly advisable.
6. The data on PDE4D promoter and exon1 methylation changes with D-galactose treatment are less than impressive. Changes are seen in only 1 site (of 6 tested) in the promoter and only 1 site (of 12 tested) in exon1, and these changes are very modest (although statistically different).
Minor Points:
1. Introduction: The limiting side effect of the current generation of PDE4-selective inhibitors is primarily diarrhea, rather than nausea/emesis. The authors need to improve this section and add better references related to the side effects in humans of the currently-available drugs, roflumilast and apremilast.
2. Methods: There are a number of errors of language and usage. In section 2.5, What is the meaning of the sentence, “Three multiple holes are made for each sample to reduce the test error.”?
3. Figure 1, panel a: “Befoer” is incorrect.
4. Figure 2, panels e and f: Please identify all 9 lanes in each immunoblot.
5. Figure 3, panel b: Please identify all 9 lanes in each immunoblot.
6. Figure 4, panel b: Please identify all 9 lanes in each immunoblot.
7. There are many places where the authors use the term PDE 4B. It should be PDE4B (no space) in all cases.
Author Response
Thank you very much for your modification suggestions. Now according to your suggestions, the author makes the following modifications
Major Points:
Point1.The authors use treatment with D-galactose as a model for the aging brain. They provide 3 references on this approach (their references 27, 28 and 29). None of these references is from a top-100 journal. There needs to be more emphasis in the Introduction and also in the Discussion on the various aspects of the model: How does it compare to study of actual aged rats, and why was it used here instead of using aged rats or mice? The use of the term “aged rats” in the title is misleading: these are not aged rats, they are D-galactose-treated rats.
Response 1:The D-galactose aging model is an internationally recognized accelerated aging model for the study of aging, used as a substitute for naturally aging rodents.Based on your suggestions, the author of the article made this change to the aging rat in the title of the article, added the relevant description of D-galactose aging model to the introduction and discussion analysis part of the article, and changed the better cited literature.The specific changes are as follows:
Title:Moderate-Intensity Intermittent Training Alters the DNA Methylation pattern of PDE4D Gene in Hippocampus to improve the ability of Spatial Learning and Memory in Aging Rats reduced by D-galactose.
Introduction: Here, in order to explore the effect of MIIT on PDE4 expression and its role in the recovery of cognitive aging, we selected the D-galactose (D-Gal)-induced aging model, which is a systematic and homogeneous aging model of comprehensively accelerated aging and cognitive deficits.
Discussion:Using senescence-accelerating model organisms is an important method to study the cellular and molecular mechanisms of hippocampal aging and cognitive decline. Long-term administration of high doses of D-galactose induces memory and cognitive impairment in rodents. Superoxide anion produced by D-galactose metabolism directly damages tissues and organs, causes oxidative damage and neuroinflammation to the brain, further damages the plasticity of synaptic structure and function of hippocampus, and finally leads to neurodegenerative symptoms .Therefore, the aging model induced by D-galactose can be used to simulate the brain aging process and evaluate the effect of anti-aging research.
Point2. In their discussion of HIIT and MIIT (and the relative virtues ofeach), the authors cite 2 references (their references 20 and 21).These references appear to focus on the cardiac and metabolic effects of these interventions, rather than their effect on the CNS. What is the state of the literature on the effects of these interventions on the aging CNS? It appears that references 30 and 31 refer to exercise paradigms other than HIIT and MIIT.
Response 2:The comparative study of MIIT and HIIT mainly focuses on the study of heart and metabolism, and the comparative study of the intervention effect on cognitive function is rare. In the original writing, the author did not make clear statement. Now, according to your suggestion, the following modifications are made:
In recent years, high-intensity interval training (HIIT) has attracted more and more attention as a time-efficient exercise. HIIT is characterized by short periods of high-intensity exercise alternating with rest or low-intensity exercise. Similar or greater mental and physical health benefits can be obtained with reduced time investment compared to traditional moderate to high intensity continuous exercise[21-23] .Studies have shown that moderate intensity interval training (MIIT) and HIIT have similar training effects in cardiac and metabolic studies [24-27], but MIIT is more suitable for the elderly or groups with low activity capacity[28], MIIT is relatively rare in cognition-related studies.
Point2.There needs to be more Methods information on the MorrisWater Maze. What instrumentation (video or other) was used tomonitor the animals (please specify manufacturer) and what software was used to capture the data, prepare statistical analyses, and prepare the Morris Water Maze figures?
Response 3:According to your revision comments, the information of water maze, camera and image acquisition system is supplemented in the article, as follows:
The water-maze (ZH, Anhui Zhenghua Biological Instrument Co., Ltd. Anhui, China);A camera(WV-CP500D, Panasonic, Japan) above the tank was used to record the swimming track of the animal. Behavioral data were recorded and analyzed by a computer installed with a visual analyzing system (ZH, Anhui Zhenghua Biological Instrument Co., Ltd. Anhui, China).
Point4. Why was only the Morris Water Maze test used, instead ofadditional tests of learning, memory and cognition?
Response 4:This study was aimed at the effects of exercise on hippocampal spatial learning and memory. The Morris water maze test is a task primarily conceived to assess the spatial learning and memory paradigm of cognitive impairments in rodents . MWM is a type of test extensively used and well-accepted for spatial learning in mice and rats.
Point5. There needs to be some additional information regarding theimmuno- blots. The authors state that PKA, PDE4A, and PDE4Dantibodies were obtained from Affinity. The authors need to provide references to prior studies that validate these antibodies.Alternatively, they need to provide data/figures that validate the antibodies: e.g., compare the signal generated in cells transfected to express PKA, PDE4A, or PDE4D, respectively, with appropriate controls (e.g., vector-only transfected cells).References on prior studies that have used the other antibodies (e.g., PDE4B, CREB, pCREB, BNDF) would be highly advisable.
Response 5: In the original writing of the author, the source information of PKA, PDE4A, or PDE4D antibody is incomplete, and the following modifications are made:
rabbit anti-PDE4A antibody (DF13317,1:500, Affinity Biosciences Cincinnati, OH, USA), rabbit anti-PDE4D antibody (DF6535,1:800; Affinity Biosciences Cincinnati,OH,USA),rabbi tanti-PKA antibody(AF5450,1:800,Affinity Biosciences Cincinnati, OH, USA), rabbit anti-BDNF antibody (DF6387 , 1:700, Affinity Biosciences Cincinnati, OH, USA). Most of the rabbit polyclonal antibodies of Affinity Biosciences have passed independent antibody validation (one of the five antibody validation strategies recommended by the International Antibody Validation Working Group), and adopt the production process of polypeptide antigen + affinity purification, with strong antibody affinity and higher specificity, meeting the internationally recognized definition of monospecific antibody.
Point6. The data on PDE4D promoter and exon1 methylation changeswith D-galactose treatment are less than impressive. Changesare seen in only 1 site (of 6 tested) in the promoter and only 1 site (of 12 tested) in exon1, and these changes are very modest (although statistically different).
Response 6 : Studies have shown that DNA methylation level at specific sites may affect the epigenetic mechanism of gene expression, such as:“A longitudinal and transancestral analysis of DNA methylation patterns and disease activity in lupus patients”\“The Genomic Impact of DNA CpG Methylation on Gene Expres- sion; Relationships in Prostate Cancer”\“Exposure to a real traffic environment impairs brain cognition in aged mice”,there are similar reports in many studies. Although there is no direct evidence to prove that the changes of PDE4D expression in aging model and exercise intervention group are related to the methylation changes of specific CG sites, there is a correlation between the changes of two CG sites and the expression of PDE4D gene in aging model and exercise intervention group rats, so exercise may regulate the expression of PDE4D gene by affecting the specific CG sites of PDE4D.
Minor Points:
Point1. Introduction: The limiting side effect of the current generationof PDE4-selective inhibitors is primarily diarrhea, rather thannausea/emesis. The authors need to improve this section andadd better references related to the side effects in humans of the currently-available drugs, roflumilast and apremilast.
Response 1 Make the following modifications according to your modification comments:
For example, inhibition of PDE4 by the first-generation inhibitor Rolipram increased the level of cAMP and p-CREB in hippocampus, as well as the survival and proliferation of new neurons in hippocampal dentate gyrus, significantly improving hippocampal long-term potentiation[10-12] . Later, several research groups reported different types of small molecule PDE4 inhibitors, such as: roflumilast, apremilast et al[13-15] . However, despite the broad therapeutic potential of these small molecule PDE4 inhibitors in preclinical studies, the progress of most molecules into the clinic has been hampered, possibly due to severe side effects such as headache, diarrhea, dizziness[7, 16,17].
Point2. Methods: There are a number of errors of language and usage. In section 2.5, What is the meaning of the sentence, “Three multiple holes are made for each sample to reduce the test error.”?
Response 2:Amend as follows: all reactions were conducted in triplicate.
Point3. Figure 1, panel a: “Befoer” is incorrect.
Response 3:Modified, see word document for details.
Point4. Figure 2, panels e and f: Please identify all 9 lanes in eachimmunoblot.
Response 4:Modified, see word document for details.
Point5. Figure 3, panel b: Please identify all 9 lanes in eachimmunoblot.
Response 5:Modified, see word document for details.
Point6. Figure 4, panel b: Please identify all 9 lanes in eachimmunoblot.
Response 6:Modified, see word document for details.
Point7. There are many places where the authors use the term PDE4B. It should be PDE4B (no space) in all cases.
Response 7:The full text has been revised
